# Comparison between School-Age Children with and without Obesity in Nutritional and Inflammation Biomarkers

**DOI:** 10.3390/jcm11236973

**Published:** 2022-11-26

**Authors:** Eias Kassem, Wasef Na’amnih, Maanit Shapira, Asher Ornoy, Khitam Muhsen

**Affiliations:** 1Department of Pediatrics, Hillel Yaffe Medical Center, Hadera 3810101, Israel; 2Rappaport Faculty of Medicine, Technion-Israel Institute of Technology, Haifa 3109601, Israel; 3Department of Epidemiology and Preventive Medicine, School of Public Health, Sackler Faculty of Medicine, Tel Aviv University, Tel Aviv 6997801, Israel; 4Laboratory Division, Hillel Yaffe Medical Center, Hadera 3810101, Israel; 5Adelson School of Medicine, Ariel University, Ariel 4077625, Israel; 6Laboratory of Teratology, Department of Medical Neurobiology, The Hebrew University Hadassah Medical School, Jerusalem 9112002, Israel

**Keywords:** iron status, lipid levels, inflammation markers, obesity, pre-adolescents

## Abstract

Childhood obesity is a major health problem. We examined differences between children with obesity and normal weight in nutritional and inflammation biomarkers. A cross-sectional study was conducted among healthy children aged 10–12 years from Arab villages in Israel. Parents were interviewed regarding sociodemographic and children’s health status. Body weight and height measurements were performed and weight categories were defined using the 2007 WHO growth curves. Blood samples were tested for complete blood count, levels of iron, ferritin, lipids, uric acid, and C-reactive protein (CRP). Overall, 146 children (59.0% males, mean age = 11.3 [SD = 0.5]) were enrolled. In total 43.8%, 14.1% and 42.3% of the participants had normal weight, overweight and obesity, respectively. A multivariable logistic regression model showed that children with overweight and obesity had lower iron, and HDL-C levels than children with normal weight. Levels of CRP, uric acid, LDL-C and lymphocytes were higher among children with overweight and obesity. In conclusion, our findings highlight the worse metabolic and nutritional status in overweight and obese children. Such markers play a role in metabolic syndrome, thus suggesting that metabolic syndrome might start in childhood.

## 1. Introduction

Childhood obesity is a major health concern [1,2]. Obesity in childhood and adolescence was shown to be significantly associated with cardiovascular disease, hypertension, type 2 diabetes mellitus, and increased risk of all-cause mortality in adulthood [3,4].

Pediatric obesity might also have an adverse impact on children’s health. For example, an inverse association was shown between the levels of serum iron and adiposity [5]. Children and adolescents might suffer from iron deficiency because their diets are usually of low quality [6,7,8]. Dyslipidemia is another health concern related to pediatric obesity. Children with obesity were shown to have a higher prevalence of dyslipidemia (28.0%) than children with normal weight (5.2%) [9]. A cohort study of 152,820 children and adolescents based on the database of the largest health maintenance organization in Israel, showed that normal-weight children had lower levels of total cholesterol, low-density lipoprotein cholesterol (LDL-C), and triglycerides and a higher level of high-density lipoprotein cholesterol (HDL-C) compared with overweight and obese children [10]. Healthy children are not routinely tested for lipids levels. It was shown that boys of the Arab ethnic minority and residents of low socioeconomic status (SES) regions had a lower likelihood to be tested for lipid profiles in Israel [11].

Moreover, obesity was shown to be associated with a low-grade inflammation that stimulates inflammatory cytokine release, including tumor necrosis factor, interleukin-6, and C-reactive protein (CRP) [12,13,14,15,16].

Despite this evidence, studies linking childhood obesity and nutritional and metabolic health status studies among metabolically healthy children with overweight or obesity are limited [17]. Moreover, most prior studies [18,19,20,21,22,23] have focused on one or two aspects, such as lipid levels and inflammation biomarkers, with limited evidence on nutritional biomarkers (e.g., iron, folate, and vitamin B12). Additionally, most studies included children with a wide range of ages spanning from childhood to late adolescence, thus mixing children from distinct developmental stages that differ significantly in hormonal profile, physical growth, and nutritional needs. Accordingly, the aim of the current study was to examine differences in multiple nutritional and inflammation biomarkers between pre-adolescents with obesity and overweight to those with normal weight, while focusing on metabolically healthy preadolescents.

## 2. Materials and Methods

### 2.1. Study Design and Population

A cross-sectional study was performed among healthy children aged 10–12 years from two Arab villages (A and B) with different SES in the Hadera sub-district in Israel. The population of Israel includes the majority (74%) of the Jewish population, and the Arab population comprises 21% of the population, while 5% belong to other groups [24]. The Arab population in Israel has lower educational levels and SES compared to the Jewish population [25]. The Arab population is characterized by higher mortality rates of cardiovascular disease and diabetes [26], with a higher prevalence of diabetes mellitus (47% vs. 34%) and smoking (57% vs. 34%) mainly in men, compared with the Jewish population [27]. The prevalence of childhood obesity increased in Israel between 2003–2004 and 2015–2016 from 10.5% to 18.2% and 6.5% to 10.5% among Arabs boys and girls aged 12–18 years, respectively. Among Jewish boys and girls of the same age, obesity prevalence increased from 8.4% to 10.7% and 5.1% to 8.6%, respectively during the same period [28]. The prevalence of obesity is higher in Arab children than in Jewish children [28,29]. We have shown a 42% prevalence of obesity among school children aged 10–12 years, which is positively associated with dietary fat intake [30]. The traditional diet in the Arab population is based on the Mediterranean diet, including large quantities of fruit and vegetables [31]. However, dietary changes were demonstrated in the Arab population, including more consumption of processed food, sweets, and soft drinks [28].

The design and study population of the current study have been described [30,32,33,34]. Briefly, participants were initially recruited as healthy babies (*n* = 233) during 2007–2008 at a median age of 8 weeks and followed up until age 18 months to document breastfeeding practices, physical growth, and acquisition of *H. pylori* infection [32]. A follow-up was conducted among the same children during 2017–2019, at ages 10–12 years, in which we obtained updated information on socio-demographics, health status, anthropometric measurements, and blood samples [30]. We established contact with 207 (88.8%) of the families, among those 189 agreed to participate in the follow-up study but 15 withdrew and 174 (84.0%) completed the questionnaire. Overall, 149 (71.9%) children of those who were successfully contacted completed the anthropometric measurements, and of those 146 performed blood tests and were included in the current study. Parents were interviewed in Arabic regarding updated sociodemographic data and children’s health status. Anthropometric measurements were performed by trained nurses and non-fasting blood samples were obtained by a specialist in pediatrics.

### 2.2. Laboratory Methods

Non-fasting blood samples of 10 mL peripheral venous blood were obtained from each child by a trained pediatrician. Blood was drawn into plain and EDTA-containing VACUETTE^®^ Blood Collection Tubes (Greiner Bio one GmbH, Kremsmünster, Austria) handled, and stored by trained staff according to accepted protocols. The blood samples were transferred in cool conditions to a central clinical diagnostic laboratory at Hillel Yaffe Medical Center within 2 h of collection. All assays were performed on the same day in a blinded manner to the demographic, clinical, and nutritional data. A complete blood count (CBC) was conducted using a DxH-800 hematology analyzer (Beckman Coulter Inc., Brea, CA, USA). Levels of biochemical markers in serum were measured using Cobas-8000 instrument with Roche Diagnostics kits according to the manufacturer’s procedures (Roche Diagnostics GmbH, Mannheim, Germany).

### 2.3. Anthropometric Data

Anthropometric measurements were performed by trained nurses. Body weight was measured to the nearest 0.1 kg using Tanita digital Scale (WB-800H plus) with light clothing but without shoes [35]. Standing height was measured with a stadiometer attached to the scale. Body mass index (BMI) was computed using Quetelet’s equation [weight (kg)/height^2^ (m^2^)]. BMI z-score was computed using the 2007 World Health Organization (WHO) population growth curves [36].

### 2.4. The Dependent Variable

Obesity was defined as a BMIZ score > 2 standard deviations (SD) for age and sex. Overweight was defined as a BMIZ score between ≥1 SD and ≤2 SD, and normal weight was defined as a BMIZ score ≥ −2 SD and <1 SD [37]. Children with overweight and obesity were grouped together and compared with children with normal weight.

### 2.5. The Independent Variables

Sociodemographic variables: age at enrollment (in years), sex, monthly household income, and the village of residence. A household density index was calculated as the number of persons living in the household divided by the number of rooms in the household.

Lipids levels: triglycerides, total cholesterol, HDL-C, and LDL-C. Dyslipidemia was determined according to the Lipid Research Clinical Prevalence Study and the United States National Health and Nutrition Examination Surveys [38,39] as having one or more of the following criteria regarding abnormal lipid levels in fasting blood samples: [1] Total cholesterol ≥ 200 mg/dL; [2] LDL-C ≥ 130 mg/dL; [3] triglycerides ≥ 130 mg/dL; [4] HDL-C ≤ 40 mg/dL among adolescence. Lipid screening can be performed with a full fasting lipid profile or with non-fasting lipid levels [40]. Results from the National Health and Nutrition Examination Surveys demonstrated only small differences between non-fasting and fasting lipid measurements that were considered as likely clinically insignificant differences [40]. Accordingly, the use of non-fasting blood samples in our study likely has limited clinical implications or significance.

Inflammation biomarkers: We used parameters of CBC platelets, leukocytes, neutrophils, and lymphocytes as measures of inflammation. Additionally, we measured serum CRP and uric acid levels.

Hemoglobin (g/dL) level (a continuous variable): was determined as part of CBC panel. Anemia was defined as hemoglobin <12 g/dL for children aged 6–14 years [41,42].

Iron deficiency was defined as serum ferritin concentrations <15 μg/dL [42,43,44].

Additional biomarkers: Protein (total; globulin and albumin), vitamin B12 and folic acid, transferrin, transferrin-saturated, mean corpuscular volume and mean corpuscular hemoglobin.

### 2.6. Statistical Methods

The assumption of the normal distribution of the study variables was tested by the Kolmogorov–Smirnov test. Normally distributed continuous variables in this study were described as the mean and standard deviations (SD), while continuous variables with skewed distributions were displayed as the median and interquartile range (IQR). Differences in sociodemographic factors and levels of iron stores, lipids, and CRP between children with and without obesity were assessed using the chi-square test for categorical variables and the Student’s *t*-test for continuous variables; the Mann–Whitney U test was used for variables with skewed distribution. Odds ratios (OR) and 95% confidence intervals (CI) were calculated for the independent variables, from multivariable logistic regression models, in which obesity/overweight was the dependent variable. Variables that were associated with obesity/overweight in bivariate analysis with *p* < 0.05 were included in the model. The final model was selected based on model fit parameters such as Nagelkerke R^2^. A sensitivity analysis was performed which compared children with obesity to children with normal weight while excluding from the analysis children with overweight. In an additional analysis, we compared study outcomes between the 3 groups of children with obesity, overweight and normal weight using the one-way ANOVA test, the Kruskal–Wallis test for variables that did not follow a normal distribution, and adjustment for multiple comparisons was done using the Bonferroni test. All statistical tests were two-sided, and *p* < 0.05 was considered statistically significant. Data analysis was performed using Statistical Package for the Social Science (SPSS) version 27 (IBM, Armonk, New York, NY, USA).

## 3. Results

Overall, 146 children (59.0% males) with a mean age of 11.3 (SD = 0.5) who performed blood tests were included in the study. Of those 65 (43.6%) had normal weight, 21 (14.1%) had overweight and 63 (42.3%) had obesity. Children with overweight and obesity were joined together and compared to children with normal weight in the subsequent analyses. There were no significant differences in age, sex, and monthly household income between the participants with overweight/obesity and those with normal weight. The median number of siblings was slightly higher among participants with overweight/obesity compared with participants with normal weight *p* = 0.03. The proportion of participants who lived in the low SES village was higher among participants with overweight/obesity vs. participants with normal weight (70.2% vs. 49.2%), *p* = 0.009 (Table 1). 

Dyslipidemia and iron deficiency were significantly more common in participants with overweight/obesity, but the difference in anemia prevalence between the groups was not significant (Figure 1).

### Associations of Nutritional and Inflammation Biomarkers with Overweight/Obesity

The levels of lymphocytes and platelets were higher among children with overweight/obesity compared with children with normal weight (*p* = 0.01; *p* = 0.005, respectively). The median level of iron was (*p* = 0.002) lower in participants with overweight/obesity (56.0 mcg/dL) than in participants with normal weight (68.0 mcg/dL), as well as the transferrin-saturated level of l 15.8% vs. 19.4%. The median HDL-C level was lower in the overweight/obese group (44.0 mg/dL) compared with the normal weight group (58.0 mg/dL); *p* < 0.001. Serum ferritin, uric acid, triglycerides, LDL-C, globulin, and CRP levels were significantly higher among the overweight/obesity group than in the normal weight group (Table 2). There were no significant differences between the groups in the levels of hemoglobin, leukocytes, neutrophils, hematocrit, mean corpuscular volume, transferrin, vitamin B12, folic acid, total cholesterol, protein, and albumin (Table 2). 

Significant differences between males and females in the levels of iron, transferrin, and ferritin, but not in other biomarkers (Appendix A).

A comparison across the 3 groups of children with obesity, overweight and normal weight showed significant differences between these groups in the levels of lymphocytes, platelets, iron parameters, vitamin B12, uric acid, triglycerides, HDL-C, LDL-C, globulin, and CRP levels (Table 3). Correction for multiple comparisons using the Bonferroni test showed significant differences between children with obesity and those with normal weight in the levels of platelets, iron parameters, uric acid, triglycerides, HDL-C, LDL-C, globulin, and CRP. Significant differences were found between participants with obesity and those with overweight in the levels of transferrin saturation, vitamin B12, HDL-C, and CRP (Table 3).

Iron and transferrin-saturated were highly correlated (Phi correlation coefficient 0.95, *p* < 0.001), and HDL-C and CRP had a moderate correlation (Phi correlation coefficient 0.44, *p* < 0.001), therefore, these variables were analyzed in separate models.

In a multivariable analysis (Model 1 Table 4), living in a low SES village compared to a high SES village was positively associated with overweight/obesity prevalence (OR 8.57, 95% CI 2.48–29.59). Negative associations were found between iron levels and overweight/obesity (adjusted OR 0.97, 95% CI 0.95–0.99), and HDL-C (OR 0.92, 95% CI 0.88–0.97) and overweight/obesity. A positive association was found between uric acid and overweight/obesity (OR 1.66, 95% CI 1.01–2.72, *p* = 0.04) in addition to LDL-C and lymphocytes (*p* = 0.004; *p* = 0.03, respectively). There were no significant differences in sex between participants with overweight/obesity and those with normal weight (OR 1.87, 95% CI 0.68–5.13, *p* = 0.2). Model 2 which included the variables “transferrin-saturated and CRP” instead of “iron and HDL-C”, respectively, showed similar results, with a significant positive association between the CRP and overweight/obesity (OR 1.39, 95% CI 1.10–1.76, *p* = 0.007) (Table 4).

A sensitivity analysis that compared participants with obesity (BMIZ scores > 2 SD) with participants with normal weight; while excluding those overweight from the analysis, showed similar results (Appendix A).

## 4. Discussion

We demonstrated that overweight and obesity among healthy participants at age 10–12 years, were positively associated with inflammation biomarkers, and worse lipid levels, in addition to lower iron levels. The prevalence of dyslipidemia in children overweight/obese in our study was 53.7%, which is consistent with previous reports [18,45]. The prevalence of dyslipidemia among obese children was estimated at 48.8% in studies from Europe and 63% in studies from South America [46]. We found higher levels of serum LDL-C and triglycerides and a lower HDL-C level in obese and overweight children as compared to the normal weight group, thus confirming findings from previous studies [9,10,19,47,48,49].

A low-grade systemic inflammation condition was detected in obese children [50]. Interestingly we found increased levels of systemic inflammation markers (CRP and lymphocytes) among metabolically healthy preadolescents with overweight and obesity in comparison to healthy participants, in line with previous studies [15,32]. Systemic inflammation is correlated with multiple adulthood diseases, including metabolic syndrome and cardiovascular disease [51,52]. Therefore, the observed associations of CRP, lymphocytes, and worse lipid profile with overweight and obesity at preadolescence are of concern and support the aggregation of multiple cardiovascular risk factors, already in childhood, several decades before the clinical onset of cardiovascular disease in adulthood. Moreover, our study highlights the role of low SES in obesity and overweight. Therefore, interventions to reduce cardiovascular risk should also target children and preadolescents, especially those from low SES settings. We found no significant difference between males and females in the prevalence of obesity and overweight. Isasi et al. in their study of Hispanic/Latino youth 8–16 years of age found that younger boys were more likely to have obesity class II-III than girls [53]. Moreover, boys were more likely to have prediabetes than girls, especially at older ages [53]. We adjusted for sex in the multivariable models, although in our data there were no significant differences between male and female children, in the levels of lipids or inflammation markers.

The association of low iron levels and iron deficiency with overweight/obesity highlights the poor nutritional status of overweight/obese children. The prevalence of iron deficiency in our study was similar to that reported among 6 to 12-year-old Thai children (32.4%) [54] but slightly lower than the reported prevalence among Israeli children and adolescents with obesity about two decades ago (38.8%) [6]. Findings from the US National Health and Nutrition Examination Survey showed a relatively low prevalence of iron deficiency among children overweight, but it was higher by almost 2-fold compared to children with normal weight [21]. A cross-sectional study from the United States demonstrated an adverse association between iron deficiency and cognitive achievement in childhood and adolescence [55].

We found a positive association between serum uric acid levels and obesity, in agreement with other reports [22,56]. A study from Taiwan showed that a high level of uric acid (≥7.3 for males and ≥6.2 mg/dL for females) increased the risk of hypertension in males and females, and metabolic syndrome in male adolescents after 10 years of follow-up [57]. Cho et al. reported that serum uric acid level is positively associated with metabolic syndrome in children [58]. Accordingly, uric acid might be a useful marker of metabolic syndrome.

Our study was conducted just before the COVID-19 pandemic, which was associated with significant changes in the daily activities of children, including school closures, virtual home-schooling, stress, increased screen time, and intermittent isolation and quarantine [59]. The impact of the COVID-19 pandemic on children’s health in general and on obesity, in particular, remains to be determined, nonetheless, it was already shown that the COVID-19 pandemic has exacerbated the obesity epidemic and widened disparities among children with selected chronic conditions [60]. A study from the USA showed that obesity prevalence increased from 23.8% among 293,341 patients aged 5 to <20 years before the pandemic to 25.5% during the pandemic, with the largest increases among children aged 5–12 years who were male, Black, or Hispanic [60]. Changes in lifestyle toward unhealthier behavior were demonstrated during the pandemic in both children and adults [61,62]. Thus, further studies are needed to assess the potential impact of the COVID-19 pandemic on obesity and the nutritional status of sensitive age groups, especially children and adolescents.

Our study had some limitations. The cross-sectional study design might limit the understanding of the causal relationships between overweight/obesity and the tested biomarkers. The small number of participants, and needs to be confirmed with a larger cohort study. Only 70.5% of the participants agreed to perform blood tests, despite multiple invitations. Our results were derived from Arab children in Israel, a population of relatively low SES, and a traditional lifestyle that is undergoing lifestyle and dietary changes towards westernized diets. Therefore, our findings can be generalizable to populations with similar characteristics.

Blood tests were performed in non-fasting conditions, which might result in a non-differential misclassification of the lipids levels. It was shown that total cholesterol, LDL-C, and HDL-C were not much different in fasting and non-fasting states; however, the fasting state is essential for the measurements of triglyceride levels because they remain high for several hours after a meal [40]. Accordingly, the interpretation of the lipid levels in our study should be made of caution.

The medical history of the study participants was collected by reports from their mothers, which might have led to non-differential misclassification.

Strengths of our study include; focusing on the preadolescents from the Arab ethnic minority and the inclusion of children from different SES villages. Usually, this population tends towards underutilization of blood tests and higher cardiovascular risk in adulthood. The study participants belonged to a well-defined cohort in terms of SES and health status. We assessed multiple biomarkers thus covering nutritional, biochemical, hematological, and metabolic aspects. Lastly, our study was based on community and conducted among healthy children aged 10–12 years, which is a homogeneous group with comparable developmental stages, physical growth, and nutritional needs.

Future studies with larger sample sizes are needed. Longitudinal studies will be essential to assess the potential effect of obesity and its related nutritional, metabolic, and inflammatory disturbances during preadolescence on the cardio-metabolic health profile in adolescence as well as the long-term effect on the development of cardiovascular and metabolic disease in adulthood.

## 5. Conclusions

Children with overweight and obesity had higher levels of inflammation markers (such as CRP and lymphocytes), worse lipid profiles (lower HDL-C, higher triglycerides, and LDL-C levels), and lower levels of serum iron compared to children with normal weight. These findings highlight the poorer metabolic and nutritional status of overweight and obese children. Such markers play a role in metabolic syndrome, thus suggesting that metabolic syndrome might start in childhood. Further studies in a larger population, preferably with interventions such as weight loss program, is warranted to clarify this association.

## Figures and Tables

**Figure 1 jcm-11-06973-f001:**
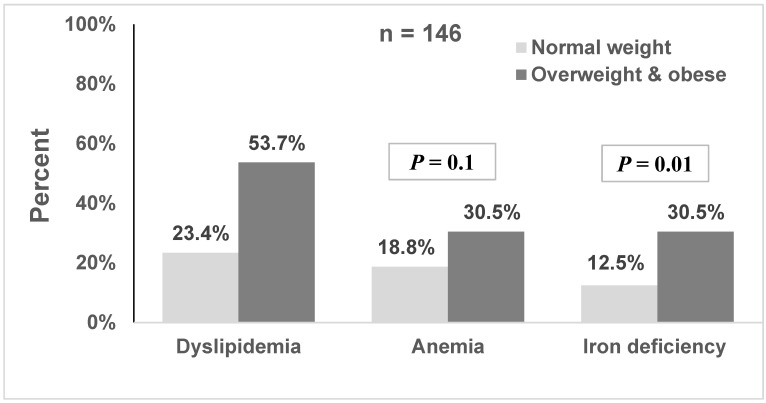
Distribution of dyslipidemia, anemia, and iron deficiency by obesity.

**Table 1 jcm-11-06973-t001:** Sociodemographic characteristics of the participants according to weight category.

	Overweight/Obesity *n* = 84	Normal Weight *n* = 65	*p*-Value ^a^
**Age, mean (SD)**	11.3 (0.5)	11.2 (0.6)	0.2 ^b^
**Sex**			0.9
**Males**	50 (59.5%)	38 (58.5%)	
**Females**	34 (40.5%)	27 (41.5%)	
**Village of residence**			0.009
**Higher SES village**	25 (29.8%)	33 (50.8%)	
**Lower SES village**	59 (70.2%)	32 (49.2%)	
**Monthly household income**			0.3
** < ** **5000 NIS**	48 (57.8%)	43 (67.2%)	
**>5000 NIS**	35 (42.2%)	21 (32.8%)	
**Mean household crowding index (SD)**	1.5 (0.6)	1.4 (0.8)	0.5 ^b^

*p* value was obtained by the ^a^ chi-square test; ^b^ Student’s *t*-test NIS: new Israeli shekel; SD: standard deviation; SES: socioeconomic status.

**Table 2 jcm-11-06973-t002:** Differences in the plasma/serum levels of nutritional and inflammatory markers between participants with normal and those with obesity/overweight.

	Overweight/Obesity *n* = 82	Normal Weight *n* = 64	*p* Value ^a^
**White blood cells (10^3^/μL), mean (SD)**	7.9 (2.4)	7.2 (2.2)	0.07
**Neutrophils (10^3^/** **μ** **L), mean (SD)**	3.8 (1.6)	3.5 (1.5)	0.2
**Lymphocytes (10^3^/μL), median (IQR)**	3.0 (1.0)	2.8 (1.0)	0.01 ^b^
**Platelets (10^3^/μL), median (IQR)**	301.0 (101.5)	282.0 (70.3)	0.005 ^b^
**Glucose (mg/dL), mean (SD)**	94.0 (12.6)	92.13 (16.8)	0.4
**Hemoglobin (g/dL), mean (SD)**	12.3 (0.9)	12.6 (0.8)	0.08
**Hematocrit (%), mean (SD)**	37.0 (2.5)	37.0 (2.3)	0.98
**Mean Corpuscular Hemoglobin (pg), median (IQR)**	26.4 (2.5)	27.3 (2.2)	0.01 ^b^
**Mean Corpuscular Volume (fL), median (IQR)**	78.9 (5.6)	79.6 (7.8)	0.6 ^b^
**Iron (mcg/dL), median (IQR)**	56.0 (28.5)	68.0 (38.8)	0.002 ^b^
**Transferrin (mg/dL), mean (SD)**	288.1 (39.2)	284.1 (34.3)	0.5
**Transferrin-saturated (%), median (IQR)**	15.8 (8.8)	19.4 (12.1)	0.002 ^b^
**Ferritin (ng/mL), median (IQR)**	49.5 (41.8)	40.0 (25.2)	0.01 ^b^
**Vitamin B12 (pg/mL), mean (SD)**	527.6 (196.2)	527.9 (202.5)	0.99
**Folic Acid (ng/mL), median (IQR)**	6.8 (4.4)	8.0 (3.2)	0.2 ^b^
**Uric Acid (mg/mL), median (IQR)**	4.3 (1.6)	3.5 (1.1)	<0.001 ^b^
**Triglycerides (mg/dL), median (IQR)**	104.5 (85.5)	84.0 (45.8)	0.001 ^b^
**Total cholesterol (mg/dL), median (IQR)**	151.5 (32.0)	147.5 (29.8)	0.3 ^b^
**HDL cholesterol (mg/dL), median (IQR)**	44.0 (12.3)	58.0 (18.0)	<0.001 ^b^
**LDL cholesterol (mg/dL), mean (SD)**	77.9 (20.3)	68.9 (17.8)	0.005
**Protein (g/L), mean (SD)**	7.4 (0.3)	7.2 (0.4)	0.1
**Albumin (g/L), median (IQR)**	4.7 (0.3)	4.7 (0.2)	0.1 ^b^
**Globulin (g/L), median (IQR)**	2.7 (0.4)	2.5 (0.4)	0.002 ^b^
**C-Reactive Protein (mg/L), median (IQR)**	2.6 (4.9)	0.3 (0.6)	<0.001 ^b^

^a^ *p* value was obtained by the Student’s *t*-test; ^b^ Mann–Whitney U test. HDL: High-density lipoprotein; IQR interquartile range; LDL: Low-density lipoprotein; SD: standard deviation.

**Table 3 jcm-11-06973-t003:** Differences in the plasma/serum levels of nutritional and inflammatory markers among children with normal weight, overweight, and obesity.

	Obesity *n* = 61	Overweight *n* = 21	Normal Weight *n* = 64	*p* Value
**White blood cells (10^3^/μL), mean (SD)**	8.1 (2.1)	7.60 (3.1)	7.2 (2.2)	0.1
**Neutrophils (10^3^/** **μ** **L), mean (SD)**	3.9 (1.4)	3.66 (2.2)	3.5 (1.5)	0.4
**Lymphocytes (10^3^/μL), median (IQR)**	3.0 (1.2)	2.9 (1.0)	2.8 (1.0)	0.02 ^a^
**Platelets (10^3^/μL), median (IQR)**	308.0 (100.0)	291.0 (107.0)	282.0 (70.3)	0.006 ^b^
**Glucose (mg/dL), mean (SD)**	93.4 (12.8)	95.7 (12.7)	92.1 (16.9)	0.6
**Hemoglobin (g/dL), mean (SD)**	12.3 (0.9)	12.3 (1.0)	12.64 (0.8)	0.2
**Hematocrit (%), mean (SD)**	37.1 (2.4)	36.59 (2.8)	37.01 (2.3)	0.7
**Mean Corpuscular Hemoglobin (pg), median (IQR)**	26.2 (2.6)	26.6 (2.3)	27.3 (2.2)	0.03 ^c^
**Mean Corpuscular Volume (fL), median (IQR)**	79.4 (5.5)	78.7 (5.2)	79.6 (7.8)	0.7
**Iron (mcg/dL), median (IQR)**	52.0 (29.0)	64.0 (32.0)	68.0 (38.8)	0.001 ^d^
**Transferrin (mg/dL), mean (SD)**	293.3 (39.2)	273.1 (36.2)	284.1 (34.3)	0.08
**Transferrin-saturated (%), median (IQR)**	15.0 (8.8)	18.9 (9.0)	19.4 (12.1)	<0.001 ^e^
**Ferritin (ng/mL), median (IQR)**	49.8 (44.5)	47.2 (36.3)	40.0 (25.2)	0.02 ^b^
**Vitamin B12 (pg/mL), mean (SD)**	485.0 (163.5)	651.3 (232.7)	527.9 (202.5)	0.004 ^f^
**Folic Acid (ng/mL), median (IQR)**	6.7 (4.2)	6.78 (4.6)	7.99 (3.2)	0.3
**Uric Acid (mg/mL), median (IQR)**	4.6 (1.8)	3.9 (1.5)	3.5 (1.1)	<0.001 ^b^
**Triglycerides (mg/dL), median (IQR)**	109.0 (59.5)	100.0 (55.0)	84.0 (45.8)	0.002 ^b^
**Total cholesterol (mg/dL), median (IQR)**	147.0 (36.5)	154.0 (27.5)	147.5 (29.8)	0.6
**HDL cholesterol (mg/dL), median (IQR)**	42.0 (11.0)	48.0 (15.0)	58.0 (18.0)	<0.001 ^g^
**LDL cholesterol (mg/dL), mean (SD)**	78.1 (22.4)	77.5 (12.9)	68.9 (17.8)	0.02 ^b^
**Protein (g/L), mean (SD)**	7.3 (0.3)	7.3 (0.3)	7.2 (0.4)	0.3
**Albumin (g/L), median (IQR)**	4.6 (0.2)	4.7 (0.3)	4.7 (0.2)	0.05 ^a^
**Globulin (g/L), median (IQR)**	2.7 (0.4)	2.6 (0.5)	2.5 (0.4)	0.005 ^b^
**C-Reactive Protein (mg/L), median (IQR)**	3.9 (5.4)	1.0 (1.6)	0.3 (0.6)	<0.001 ^h^

HDL: High-density lipoprotein; IQR: interquartile range; LDL: Low-density lipoprotein; SD: standard deviation. ^a^ Post hoc pairwise comparison: no significant difference between the groups. ^b^ Post hoc pairwise comparison: obesity vs. normal weight (*p* < 0.05); no significant difference between the other groups. ^c^ Post hoc pairwise comparison of obesity vs. normal weight (*p* = 0.05); no significant difference between the other groups. ^d^ Post hoc pairwise comparison: obesity vs. normal weight (*p* = 0.001); obesity vs. overweight (*p* = 0.05); normal weight vs. overweight (*p* = 1.00). ^e^ Post hoc pairwise comparison: obesity vs. normal weight (*p* = 0.001); obesity vs. overweight (*p* = 0.01); normal weight vs. overweight (*p* = 1.00). ^f^ Post hoc pairwise comparison: obesity vs. normal weight (*p* = 0.6); obesity vs. overweight (*p* = 0.002); normal weight vs. overweight (*p* = 0.04). ^g^ Post hoc pairwise comparison: obesity vs. normal weight (*p* < 0.001); obesity vs. overweight (*p* = 0.02); normal weight vs. overweight (*p* = 0.05). ^h^ Post hoc pairwise comparison: obesity vs. normal weight (*p* < 0.001); obesity vs. overweight (*p* = 0.001); normal weight vs. overweight (*p* = 1.00).

**Table 4 jcm-11-06973-t004:** Associations of nutritional and inflammation biomarkers with overweight/obesity among children aged 10–12 years.

	Model 1 Adjusted OR (95% CI)	*p* Value ^a^	Model 2 Adjusted OR (95% CI)	*p* Value ^b^
**Village**				
**Lower SES village**	8.57 (2.48–29.59)	0.001	6.44 (1.96–21.09)	0.002
**Higher SES village**	Reference		Reference	
**SEX**				
**Males**	1.87 (0.68–5.13)	0.2	2.26 (0.86–5.93)	0.1
**Females**	Reference		Reference	
**Lymphocytes (10^3^/μL)**	2.00 (1.07–3.75)	0.03	2.10 (1.18–3.96)	0.02
**Platelets (10^3^/μL)**	1.00 (0.99–1.01)	0.7	1.002 (0.99–1.01)	0.6
**Iron (mcg/dL)**	0.97 (0.95–0.99)	0.004	Not included	
**Transferrin-saturated (%)**	Not included		0.95 (0.89–1.01)	0.1
**Ferritin (ng/mL)**	1.02 (0.99–1.04)	0.1	1.02 (0.99–1.04)	0.06
**Uric Acid (mg/mL)**	1.66 (1.01–2.72)	0.04	2.01 (1.22–3.29)	0.006
**Triglycerides (mg/dL)**	1.01 (0.99–1.02)	0.08	1.01 (1.003–1.02)	0.01
**LDL cholesterol (mg/dL)**	1.04 (1.01–1.07)	0.004	1.03 (1.003–1.06)	0.03
**HDL cholesterol (mg/dL)**	0.92 (0.88–0.97)	0.001	Not included	
**C-Reactive Protein (mg/L)**	Not included		1.39 (1.10–1.76)	0.007

CI: confidence interval; HDL: High-density lipoprotein; LDL: Low-density lipoprotein; OR: odds ratio; SES: socioeconomic status. Included in the multivariable model were 146 children (82 with overweight or obesity). *p* value ^a^ from multivariable logistic regression model 1, Nagelkerke R^2^ = 0.59. *p* value ^b^ from multivariable logistic regression model 2, Nagelkerke R^2^ = 0.57.

## Data Availability

Individual-level data from this study cannot be made publicly available due to legal restrictions.

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
