# Peer review of "Comparison between School-Age Children with and without Obesity in Nutritional and Inflammation Biomarkers"

_jcm, 2022, doi:10.3390/jcm11236973_

Round 1

Reviewer 1 Report

This study aimed to compare the metabolic markers and nutritional status in children with obesity and without obesity.

The topic in this field has been widely investigated. In this study, the authors have already distinguished nutritional status based on sex (as expected). Next to that, multivariate analysis has been performed to differ metabolic markers such as inflammation (CRP), lipid markers and other biomarkers as described in The Table 4. However, the authors have not adjusted metabolic biomarkers by sex.

It has been shown by Isasi et al. 2016 (https://pubmed.ncbi.nlm.nih.gov/27344220/), there is sex differences on metabolic biomarkers in children.

Perhaps the authors can add this analysis where sex is included as adjusted factor and what is the outcome? Please add in the discussion as well and the final conclusion.

Author Response

                                                                                                November 20, 2022

To Prof. Emmanuel Andrès

Journal of Clinical Medicine

[email protected]

Dear Prof. Emmanuel Andrès,

Re: "Comparison between school-age children with and without obesity in nutritional and inflammation biomarkers"

We are pleased that our manuscript was well accepted by the reviewers and editor and that it will be further considered for publication in the Journal of Clinical Medicine, pending our addressing the requested revisions. Accordingly, attached please find a revised version of our manuscript that addresses the comments raised by the reviewers and editor.

Below please see our responses to each of the comments.

The changes in the text are marked in colour.

We thank the reviewers for their comments and suggestions, which we believe have improved the manuscript. 

Sincerely,

Khitam Muhsen, Ph.D.

Department of Epidemiology and Preventive Medicine

School of Public Health, Sackler Faculty of Medicine, Tel Aviv University

Ramat Aviv, Tel Aviv 69978

Israel.

Tel: +972-3-6405945 Fax: +972-3-6409868

Point-by-point reply

We thank the reviewers for their valuable comments and suggestions

Reviewer 1

Comments and Suggestions for Authors

-This study aimed to compare the metabolic markers and nutritional status in children with obesity and without obesity. The topic in this field has been widely investigated. In this study, the authors have already distinguished nutritional status based on sex (as expected).

 Reply: We thank the reviewer for the positive comment

 -Next to that, multivariate analysis has been performed to differ metabolic markers such as inflammation (CRP), lipid markers and other biomarkers as described in The Table 4. However, the authors have not adjusted metabolic biomarkers by sex. It has been shown by Isasi et al. 2016 (https://pubmed.ncbi.nlm.nih.gov/27344220/), there is sex differences on metabolic biomarkers in children. Perhaps the authors can add this analysis where sex is included as adjusted factor and what is the outcome? Please add in the discussion as well and the final conclusion.

 Reply: Following the reviewer's suggestion, we modified the multivariable analysis and included sex as a covariate. Please see the revised models 1 and 2 in Table 4 and the supplementary Table S2. ”There were no significant differences in sex between the participants with overweight/obesity and those with normal weight (OR 1.87, 95% CI 0.68–5.13, P=0.2)” (Please see the results section on page 7, lines 241-243). We made an additional analysis, in which sex was the dependent variable to explore associations of nutritional and inflammation biomarkers with sex. Please see page 6, lines 203-206, and Table S3 on the supplementary file.

We also made a comment on this point in the discussion section. “We found no significant difference between males and females in the prevalence of obesity and overweight. Isasi et al. in their study of Hispanic/Latino youth 8-16 years of age found that younger boys were more likely to have obesity class II-III than girls [1]. Moreover, boys were to have prediabetes more than girls, especially at older ages [1]. We adjusted for sex in the multivariable models, although in our data there were no significant differences between male and female children, in the levels of lipids or inflammation markers.” (Please see the discussion section on page 8, lines 278-284).

References

  1. Isasi, C.R.; Parrinello, C.M.; Ayala, G.X.; Delamater, A.M.; Perreira, K.M.; Daviglus, M.L.; Elder, J.P.; Marchante, A.N.; Bangdiwala, S.I.; Van-Horn, L.; et al. Sex Differences in Cardiometabolic Risk Factors among Hispanic/Latino Youth. J Pediatr. 2016, 176, 121-127.e1. https://doi:org/10.1016/j.jpeds.2016.05.037.

Reviewer 2 Report

In this manuscript Kassem et al examined differences in multiple nutritional and inflammation biomarkers between pre-adolescents with obesity and overweight to those with normal weight, while focusing on metabolically healthy individuals. This is a follow up study from previously published reports [refs: 30, 32-34] conducted among healthy children aged 10-12 years from Arab villages in Israel.

Authors demonstrated that overweight and obesity among healthy participants at age 10- 12 years, were positively associated with inflammation biomarkers, and worse lipid levels, in addition to lower iron levels. CRP, uric acid, LDL-C and lymphocytes were higher among children with overweight and obesity.

 Authors concluded that these findings highlight the worse metabolic and nutritional status in overweight and obese children. Since such markers play a role in metabolic syndrome, they suggested that metabolic syndrome might start in childhood.

 Overall, the study is interesting and authors highlighted its novelty, stating that the most prior studies have focused on one or two aspects, such as lipid levels and inflammation biomarkers, with limited evidence on nutritional biomarkers (e.g., iron, folate, and vitamin B12). Limitations are discussed, including that blood tests were performed in non-fasting conditions, therefore proper interpretations should be done. Strengths of this study include focusing on the preadolescents from the Arab ethnic minority and the inclusion of children from different villages. Statistical analysis seems to be well described and performed thoroughly.

Author Response

Response to Reviewer 2 Comments

 Point-by-point reply

We thank the reviewer for the valuable comments and suggestions, which we believe have improved the manuscript.   

In this manuscript Kassem et al examined differences in multiple nutritional and inflammation biomarkers between pre-adolescents with obesity and overweight to those with normal weight, while focusing on metabolically healthy individuals. This is a follow up study from previously published reports [refs: 30, 32-34] conducted among healthy children aged 10-12 years from Arab villages in Israel. Authors demonstrated that overweight and obesity among healthy participants at age 10- 12 years, were positively associated with inflammation biomarkers, and worse lipid levels, in addition to lower iron levels. CRP, uric acid, LDL-C and lymphocytes were higher among children with overweight and obesity. Authors concluded that these findings highlight the worse metabolic and nutritional status in overweight and obese children. Since such markers play a role in metabolic syndrome, they suggested that metabolic syndrome might start in childhood.

Overall, the study is interesting and authors highlighted its novelty, stating that the most prior studies have focused on one or two aspects, such as lipid levels and inflammation biomarkers, with limited evidence on nutritional biomarkers (e.g., iron, folate, and vitamin B12).

Reply: We thank the reviewer for the positive comments.

- Limitations are discussed, including that blood tests were performed in non-fasting conditions, therefore proper interpretations should be done. 

Reply: Following the reviewer's comment, we emphasized the need for cautious interpretation of the lipids levels results in our study. Please see page 9, lines 325-329.

- Strengths of this study include focusing on the preadolescents from the Arab ethnic minority and the inclusion of children from different villages. Statistical analysis seems to be well described and performed thoroughly.

Reply: We thank the reviewer for the positive comment.

Sincerely,

Khitam Muhsen, Ph.D.

Department of Epidemiology and Preventive Medicine

School of Public Health, Sackler Faculty of Medicine, Tel Aviv University

Ramat Aviv, Tel Aviv 69978

Israel.

Tel: +972-3-6405945 Fax: +972-3-6409868
